# Targeted Elimination of Influenza Virus and Infected Cells with Near-Infrared Antiviral Photoimmunotherapy (NIR-AVPIT)

**DOI:** 10.3390/pharmaceutics17020173

**Published:** 2025-01-28

**Authors:** Terumi Mizukoshi, Koichiro Tateishi, Mizuki Tokusanai, Yoshiyuki Yoshinaka, Aisaku Yamamoto, Naoki Yamamoto, Norio Yamamoto

**Affiliations:** 1Medical Corporation Koujunkai, Kawasaki 211-0063, Japan; momo.mizukoshi@nifty.com (T.M.); kosugihifuka@gmail.com (A.Y.); 2Department of Microbiology, Tokai University School of Medicine, Isehara 259-1193, Japan; tateishi.koichiro.f@tokai.ac.jp (K.T.); 2cmud015@mail.u-tokai.ac.jp (M.T.); twoyoshi@outlook.com (Y.Y.); 3Genome Medical Sciences Project, National Center for Global Health and Medicine, Ichikawa 272-8516, Japan

**Keywords:** influenza virus, hemagglutinin, near-infrared, IR-700, monoclonal antibody

## Abstract

**Background**: Seasonal influenza causes significant morbidity and mortality each year. Since viruses can easily acquire drug-resistant mutations, it is necessary to develop new antiviral strategies with different targets. Near-infrared photoimmunotherapy (NIR-PIT) is a type of anti-cancer therapy that has recently attracted considerable attention, with favorable outcomes reported for several cancers. In this study, we investigated whether this approach could be used as a novel anti-influenza therapy to destroy influenza virus and infected cells. **Methods**: To evaluate the efficacy of near-infrared antiviral photoimmunotherapy (NIR-AVPIT), we prepared an anti-hemagglutinin (HA) monoclonal antibody without neutralizing activity against influenza A virus (FluV) labeled with IR-700 and reacted it with FluV and infected cells, as well as HA-expressing HEK293 cells. **Results**: NIR-AVPIT strongly inactivated FluV virions, suppressed cytopathic effects, and achieved more than a 4-log reduction in viral RNA amplification. Treatment of FluV-infected cells with the antibody-IR700 complex and NIR in the early stages of infection significantly inhibited viral propagation, and double treatment with time apart exerted a greater inhibitory effect. NIR-AVPIT rapidly induced morphological changes in HA-expressing HEK293 cells and inhibited the proliferation of these cells. **Conclusions**: These results suggest that NIR-AVPIT targeting HA antigens could inactivate FluV and eliminate infected cells in vitro. This strategy is a promising approach to treat various viral infections, including influenza.

## 1. Introduction

Approximately a billion people worldwide are infected with the influenza virus (FluV) each year, of which 3–5 million people become seriously ill and 290,000–650,000 people die from the associated respiratory symptoms [1,2]. The highest mortality rates were estimated in sub-Saharan Africa, southeast Asia, and among people aged 75 years or older [1] (Iuliano et al., 2018). Antiviral drugs targeting proteins and enzymes encoded by viral genes (e.g., M2 protein, neuraminidase, RNA polymerase, and cap-dependent endonuclease) are available for the treatment of FluV infection but are often ineffective if not used within approximately 48 h after symptom onset [3]. Studies have also reported that the early resolution of symptoms with antiviral drugs does not provide sufficient immunity [4]. Furthermore, many protective antibodies against FluV are known to bind to hemagglutinin (HA) on the surface of the virus [5,6,7,8], but the virus readily escapes from neutralizing antibodies through mutations, making it difficult to develop effective antibody therapies. Therefore, new approaches for treating FluV infections are in high demand. 

Near-infrared photoimmunotherapy (NIR-PIT) is a new cancer therapy in which a specific antibody conjugated to the sensitizer IRDye700DX (IR700) is administered to target cells. NIR irradiation induces a structural change in the antibody-IR700 conjugate and causes mechanical stress at the antibody binding site, resulting in cell death due to cytoplasmic leakage [9,10,11,12,13]. Importantly, the killing of target cells with NIR-PIT is specific, and cells without the antibody-IR700 conjugate are not affected. Moreover, no severe adverse effects have been reported in the clinical use of NIR-PIT. Therefore, NIR-PIT is expected to be a potential new strategy for the treatment of cancer. 

NIR can selectively damage cancer cells expressing a specific antigen by administering antibody-IR700 complexes. However, infectious diseases may be more suitable targets for this technology than cancers due to the pathologically high expression of the foreign antigens of the causative organisms in patients with infectious diseases. In fact, photoimmunotherapy has begun to be applied to viral infections [14,15,16], but what factors influence its efficacy have not yet been fully analyzed. In this study, we developed a new antiviral strategy called near-infrared antiviral photoimmunotherapy (NIR-AVPIT) and selected the influenza A (H1N1)pdm09 virus, which caused a pandemic in 2009 [17,18,19,20], as the first target to test NIR-AVPIT. The effects of NIR-AVPIT on free virion and FluV-infected cells as well as the relationship between antiviral efficacy and time of post-infection were evaluated.

## 2. Materials and Methods

### 2.1. Cells

Madin–Darby Canine Kidney (MDCK) and HEK293 cells were purchased from American Type Culture Collection (ATCC; Rockville, MD, USA). MDCK and HA-expressing HEK 293 cells were cultured in Dulbecco’s Modified Eagle’s Medium (DMEM; Thermo Fisher Scientific, Waltham, MA, USA) supplemented with 1 g/L D-glucose, 4 mM L-glutamine, 10% fetal bovine serum (FBS; Thermo Fisher Scientific), and 1% penicillin/streptomycin (Thermo Fisher Scientific). All cells were grown in a humidified atmosphere of 5% CO_2_ at 37 °C.

### 2.2. Plasmid Construction

A codon-optimized HA of influenza virus A/California/7/2009 (H1N1pdm09, GISAID EpiFlu database, EPI_ISL_31553) was synthesized by Eurofins Genomics (Tokyo, Japan). The PCR-amplified fragment of the HA-coding region was inserted into a pBApo-EF1α-puro vector (Takara Bio, Shiga, Japan) using an In-Fusion HD cloning kit (Takara Bio). The constructed plasmid, pBApo-EF1α-HA-puro, was used for the expression of HA in HEK293 cells.

### 2.3. Generation of Monoclonal Antibodies

The influenza virus A/California/7/2009 was propagated in embryonated chicken eggs and purified using sucrose gradient (10%–50%) ultracentrifugation. To prepare the ether-split antigen, the purified virus was mixed with an equal volume of ether and incubated at room temperature for 1 h with stirring. The mixture was centrifuged at 3000 rpm for 15 min to separate the aqueous and ether phases and the aqueous phase was collected. The HA antigen was obtained after two rounds of purification. Following a one-week acclimatization period, three 8-week-old female B6D2F1 mice were immunized with purified HA proteins. To increase the success rate of obtaining monoclonal antibodies, three mice were used for immunization. Splenocytes of the mice were fused with myeloma cells 4 weeks after immunization, and 800 wells of hybridoma cells were seeded. The supernatants were screened for reactivity using ELISA with HA antigen as the target and BSA as a negative control. Wells were considered positive if the OD difference between HA and BSA exceeded 0.05. Finally, three hybridoma cell clones were established to produce monoclonal antibodies (mAb1, mAb2, and mAb3) reactive to the HA protein. The culture medium of each hybridoma clone was centrifuged at 7000 rpm for 20 min and the antibodies were purified using AcroSep Hyper DF columns (Pall Corporation, Port Washington, NY, USA). The concentrations of purified monoclonal antibodies were measured using a BCA protein assay kit (Thermo Fisher Scientific). The immunoglobulins were characterized using a mouse immunoglobulin isotyping ELISA kit (BD Pharmingen, Franklin Lakes, NJ, USA).

### 2.4. Selection of the Anti-HA Monoclonal Antibody Without Neutralizing Activity

The three monoclonal antibodies at 1 µg/mL and 10 µg/mL concentrations and FluV (2 × 10^4^ pfu/mL) were incubated at room temperature for 1 h, and 100 µL of the mixture was added to MDCK cells in a 96-well plate. The cells were incubated at 37 °C for 48 h, and cell morphology was observed under an optical microscope CKX31 (Olympus, Tokyo, Japan). The amount of viral RNA in the culture supernatant at 48 h post-infection (hpi) was determined using RT-qPCR (reverse transcription quantitative PCR). 

### 2.5. Generation of HA-Expressing HEK293 Cells

The pBApo-EF1α-HA-puro vector was digested with ScaI (New England Biolabs, Ipswich, MA, USA) and HEK293 cells were transfected with linearized pBApo-EF1α-HA-puro using Lipofectamine 2000 (Thermo Fisher Scientific). Puromycin-resistant cells were cloned using the limiting-dilution method and clones with high cell-surface expression of HA were identified by fluorescence-activated cell sorting (FACS) analysis with an anti-HA antibody (mAb3) and goat anti-mouse IgG antibody-Alexa Fluor 488 (Abcam, Cambridge, UK) using a BD FACS Calibur Flow Cytometer (BD Biosciences, Franklin Lakes, NJ, USA). 

### 2.6. Preparation of IR700-Conjugated Antibody

mAb3 was conjugated to IR700 using an IRDye 700DX protein labeling kit (LI-COR, Lincoln, NE, USA) following the manufacturer’s instructions. Briefly, 100 µg mAb3 was mixed with 0.7 µL IR-dye700 in 0.1 mol/L K_2_HPO_4_ (pH 8.5) and incubated at room temperature for 2 h. The dye-labeled antibodies were purified using Pierce Zeba Desalting Spin Columns (Thermo Fisher Scientific). The dye and protein concentrations were measured at 689 and 280 nm, respectively, using a spectrophotometer (NanoDrop 2000c; Thermo Fisher Scientific). The dye/mAb3 ratio was estimated to be 1.5:1.

### 2.7. NIR Irradiation of FluV

The mixtures containing influenza virus A/Narita/1/2009 (2 × 10^4^ pfu/mL) and mAb3-IR700 conjugates (1 µg/mL) were incubated at 37 °C for 1 h and then irradiated with NIR at 60 J/cm^2^ using near-infrared equipment (LED S6775, Hamamatsu Photonics, Shizuoka, Japan) for 1 h. Then, 100 µL NIR-irradiated or control mixtures were added to each well of MDCK cells cultured in a 96-well plate. The cells were incubated with the mixtures at 37 °C for 1 h before the culture media containing virus and IR700-labeled mAb3 were exchanged with 100 µL Opti-PRO SFM (Thermo Fisher Scientific) with 4 mM L-glutamine and 2 µg/mL acetylated trypsin. The supernatants were collected for viral quantification after 48 h of incubation. 

### 2.8. NIR-Irradiation of HA-Expressing HEK293 Cells

HEK293 cells with high HA expression were treated with mAb3-IR700 conjugates (1 µg/mL) at 37 °C for 1 h and irradiated with NIR light at 60 J/cm^2^ for 1 h. The cell morphology was observed by microscopy 1.5 h and 24 h post-irradiation. 

### 2.9. NIR-Irradiation of Virus-Infected Cells

MDCK cells were seeded in 96-well plates and infected with FluV at 0.1 MOI. At 6 and/or 24 hpi, infected cells were incubated with 1 µg/mL mAb3-IR700 conjugates at 37 °C for 1 h. The cells were irradiated with NIR at 60 J/cm^2^ for 1 h and the culture media was replaced with Opti-PRO SFM medium with 4 mM L-glutamine and 2 µg/mL acetylated trypsin. Samples were collected for virus quantification and MDCK cells were observed under a microscope at 24 and/or 48 hpi.

### 2.10. Evaluation of Cell Viability

The viability of MDCK cells after NIR irradiation was evaluated using the MTS assay kit (CellTiter 96 AQueous One Solution Cell Proliferation Assay; Promega, Madison, WI, USA). Briefly, 20 µL MTS assay solution was added to each well of a 96-well plate containing MDCK cells 48 h after virus inoculation. The cells were incubated for 4 h at 37 °C before the optical density was measured at 490 nm (OD_490_) using a Spark M10 microplate reader (Tecan, Männedorf, Switzerland). 

### 2.11. RNA Extraction and RT-qPCR

Viral RNA was extracted using TRI reagent (Molecular Research Center, Inc., Cincinnati, OH, USA) following the manufacturer’s instructions. Culture supernatants were collected 24 or 48 h after virus inoculation and mixed with 0.9 mL TRI reagent, including total RNA from uninfected A549 cells as the loading control for normalization. Viral RNA was quantified using the TaqMan Fast Virus 1-step real-time RT-PCR assay kit (Thermo Fisher Scientific) and Quant Studio 3 real-time PCR system (Thermo Fisher Scientific), as described previously [21]. The sequences of the primer and probe sets were as follows: FluV-F, 5′-CACCTGATATTGTGGATTACTGATCG-3′; FluV-R, 5′-CACTCTGCTGTTCCTGTTGATATTC-3′; FluV-P, 5′-FAM-CCTCATGGACTCAGGCACTCCTTCCG-TAMRA-3′; 18S-F, 5′-GTAACCCGTTGAACCCCATT-3′; 18S-R, 5′-CCATCCAATCGGTAGTAGCG-3′; 18S-P, 5′-FAM-TGCGTTGATTAAGTCCCTGCCCTTTGTA-TAMRA-3′.

### 2.12. Statistical Analysis

Values are expressed as the mean  ±  standard deviation (SD). Paired Student’s *t*-tests were used to assess statistical differences and *p*  <  0.05 was considered statistically significant.

## 3. Results

### 3.1. Selection of the Anti-Influenza Virus HA Antibody for NIR-AVPIT

To create anti-HA antibodies conjugated with IR700 for use in NIR-AVPIT, we tested the neutralizing activity of the three monoclonal antibodies generated (mAb1, mAb2, and mAb3) against HA of FluV. To confirm the virucidal activity of NIR-AVPIT, an HA-specific antibody that lacked neutralizing activity was required. After MDCK cells were infected with FluV in the presence or absence of monoclonal antibodies, a small or no cytopathic effect (CPE) was observed with 10 µg/mL of mAb1 and mAb2, whereas a strong CPE was observed with mAb3 (Figure 1A). The amount of viral RNA in the culture supernatant was quantified using RT-qPCR, which confirmed that mAb1 and mAb2 significantly inhibited FluV amplification, while mAb3 did not (Figure 1B). These results indicated that mAb3 had no neutralizing activity and it is suitable for evaluation of NIR-AVPIT. Based on these results, 1 µg/mL mAb3 was used in subsequent experiments.

### 3.2. Direct Inactivation of FluV Virions by NIR-AVPIT

To evaluate the direct impact of NIR-AVPIT on FluV virions, we inoculated MDCK cells with FluV treated with mAb3-IR700, NIR, or a combination of both, and FluV infection was monitored through CPE analysis and viral RNA quantification by RT-qPCR (Figure 2A). FluV with mAb3-IR700 or NIR did not inhibit CPE progression, but the combination of mAb3-IR700 and NIR blocked FluV-induced CPE in MDCK cells (Figure 2B). The amount of viral RNA was significantly decreased after treatment with both mAb3-IR700 and NIR when compared to that of the control at 48 hpi (Figure 2C). Treatment of FluV with either mAb3-IR700 or NIR did not significantly affect FluV infection. These results demonstrated that NIR-AVPIT directly inactivated FluV virions and prevented FluV infection in MDCK cells. 

### 3.3. Effect of NIR-AVPIT Treatment on Virus-Infected Cells and HA-Expressing HEK293 Cells

Our initial experiments showed that NIR-AVPIT directly damaged FluV. Therefore, we next investigated whether NIR-AVPIT treatment was effective in FluV-infected cells. Since influenza viruses bud from infected cells approximately 6 to 8 h after infection [22], FluV-infected MDCK cells were treated with mAb3-IR700 and/or NIR either once (at 7 or 25 hpi) or twice (at 7 and 25 hpi) to evaluate their antiviral effects (Figure 3A). Cell morphology was observed by optical microscopy at 48 h after virus inoculation. The cells treated with NIR-AVPIT once at 7 hpi and twice at 7 and 25 hpi displayed clear inhibition of CPE at 48 hpi (Figure 3B). In contrast, a strong CPE was observed in cells subjected to NIR-AVPIT once at 25 hpi. 

Similarly to the results of CPE analysis, viral RNA levels were significantly reduced in the cells treated once (at 7 hpi) with NIR-AVPIT at 24 hpi, and in the cells treated once (at 7 hpi) or twice (at 7 and 25 hpi) with NIR-AVPIT at 48 hpi, when compared with the control cells (Figure 3C). Moreover, double NIR-AVPIT applications resulted in a significantly stronger inhibition of viral production than a single application of NIR-AVPIT (Figure 3C). We measured cell viability 48 h after viral inoculation using an MTS assay. Cells with single (at 7 hpi) or double (at 7 and 25 hpi) NIR-AVPIT treatments showed significantly higher viability than the FluV-infected control and showed no reduction in cell viability when compared with the negative control (Figure 3D). These results indicated that FluV-infected cells were efficiently removed by the combination of mAb3 and NIR irradiation. The effect of NIR-AVPIT on infected cells was lower than that of direct virion inactivation; however, in the early stages of viral infection, NIR-AVPIT was effective in destroying infected cells and inhibiting virus propagation.

To confirm that NIR-AVPIT specifically targeted the FluV HA protein, HEK293 cell clones expressing HA were generated. FACS analysis indicated that 293-HA clone 3 showed the highest HA expression on the cell surface (Figure 3E). The 293-HA clone 3 cells were then subjected to mAb3-IR700 treatment and NIR irradiation, followed by microscopic examination of their morphology (Figure 3F). NIR-AVPIT of 293-HA clone 3 cells induced cell rounding 1.5 h after irradiation and reduced cell proliferation for 24 h when compared to that in the other groups (Figure 3F). These results demonstrated that NIR treatment with mAb3-IR700 targeted HA proteins and damaged cells that highly expressed these proteins.

## 4. Discussion

NIR-PIT has been primarily evaluated as a therapy for various tumors, including glioblastoma, breast carcinoma, lung carcinoma, and metastases [23,24,25,26,27,28,29,30,31,32,33,34]. In the present study, we examined whether NIR-PIT could also be used as a therapy for viral infections. Vaccines and antivirals are the primary measures to combat influenza, COVID-19, and other common viral infections, but their development is fraught with challenges. For example, it generally takes considerable time for vaccines and antivirals to become commercially available, and antigenically drifted or drug-resistant strains of the virus can easily emerge, resulting in a loss of their efficacy. Thus, rapid and effective countermeasures against viral diseases are urgently needed. In this context, NIR-AVPIT targeting a highly conserved epitope among influenza viruses could be a promising solution to address these challenges.

Neutralizing antibodies can be useful against viral infections [35,36]. However, to demonstrate the effectiveness of NIR-AVPIT, it is essential to perform NIR-AVPIT using an antibody that lacks neutralizing activity. Therefore, in this study, we first developed an antibody that binds to the virus but has no neutralizing activity. Our results showed that among the three antibodies developed, mAb3 exhibited minimal to no neutralizing activity against FluV, leading to its selection for use at low concentration in subsequent experiments (Figure 1). 

Next, NIR-AVPIT using mAb3 was evaluated in FluV particles as well as in virus-infected cells. The results showed that FluV virions were strongly inactivated when treated directly with NIR-AVPIT, suggesting its utility in treating FluV (Figure 2). Furthermore, NIR-AVPIT significantly inhibited FluV amplification and FluV-induced CPE in infected cells and specifically damaged HA-expressing HEK293 cells (Figure 3). These results clearly indicated that NIR-AVPIT with mAb3 could eliminate viruses and virus-infected cells. 

Regarding the relationship between time and antiviral effects, early treatment of infected cells with NIR-AVPIT at 7 hpi resulted in significant inhibition of FluV propagation, whereas treatment at 25 hpi showed no such effect (Figure 3B–D). These results suggest that in vitro, the inhibitory effect of NIR-AVPIT on infected cells can only be achieved in the early stages of infection. In addition, NIR-AVPIT treatment had a weaker inhibitory effect against FluV-infected cells than against virions (Figure 2 and Figure 3). One possible explanation for this difference could be the time-dependent expression of viral HA antigens on the surface of infected cells. Influenza A viruses begin to produce viral antigens on the cell surface approximately 6 h after inoculation [37]. Therefore, NIR treatment applied at approximately 7 h after infection destroyed most infected cells, but some of the infected cells with delayed HA expression would evade antibody-binding and NIR-induced cell death. These results suggest that multiple NIR-AVPIT treatments are necessary to completely remove infected cells, including those with delayed HA expression. Indeed, in the present study, the amount of virus in the group that received two NIR-AVPIT treatments at 7 and 25 hpi was significantly reduced when compared to the amount of virus in cells that received a single treatment at 7 h post-inoculation (Figure 3C). 

Concerning the specificity of mAb3-IR700 for infected cells in comparison to healthy cells, it is considered to be notably higher for infected cells. As shown in Figure 3E, mAb3 did not recognize control HEK293 cells but specifically targeted HA-expressing cells. Moreover, the data in Figure 3B,D demonstrated that NIR-AVPIT selectively killed FluV-infected cells while leaving uninfected cells unharmed. In assessing NIR-PIT as a therapy for cancer, it was confirmed that phototoxicity was target-specific [10]. HER2 positive cells were specifically killed by anti-HER2-antibody-IR700 with NIR irradiation, but HER2 negative cells were unaffected. In addition, adverse events were minimal in NIR-PIT for oropharyngeal cancer [38]. Our results and their findings support the specificity of NIR-AVPIT.

The breadth of the antibody employed in NIR-AVPIT significantly influences its effective range. Infection or vaccination chiefly induces strain-specific antibodies for the globular head of HA, but these antibodies may lose effectiveness when the virus undergoes antigenic changes in the HA head under selective pressure [39]. Unlike most epitopes on the HA head, the HA stem is less permissive of amino acid substitutions and is significantly more conserved across subtypes [39,40]. Using an antibody targeting a highly conserved region for NIR-AVPIT would broaden the range of targets and enhance robustness to viral mutations. In addition, a broad-spectrum antibody would enable a reduction in the valency of NIR-AVPIT for influenza virus subtypes A/H1, A/H3, and type B. With respect to mAb3, its binding site on HA remains undetermined. Future studies are planned to investigate the epitope and assess the cross-reactivity of mAb3.

NIR-AVPIT and existing antivirals have different mechanisms to inhibit the propagation of influenza virus. A neuraminidase inhibitor such as oseltamivir blocks the release of virions from infected cells [41], and a cap-dependent endonuclease inhibitor baloxavir acid inhibits viral RNA transcription [42]. In NIR-AVPIT, when virions or infected cells are irradiated with NIR light, the axial ligands attached to the phthalocyanine dye IR700 dissociate by hydrolysis [15]. The accumulation of such events on virions and cell membranes may lead to a loss of membrane integrity and photoimmunotherapy-associated bursting, resulting in virion destruction and cell death, making the NIR-AVPIT-based antiviral approach unique and unlike existing antiviral therapies. We consider the advantage of NIR-AVPIT over existing antivirals to be its ability to directly destroy released viruses themselves. However, a potential limitation of NIR-AVPIT appears to be the relative difficulty of NIR irradiation. Without a specialized device for NIR treatment, patients are unable to perform NIR-AVPIT at home. Since NIR-AVPIT has a mechanism of action distinct from currently approved treatments, the combination of NIR-AVPIT with existing anti-influenza drugs would be expected to exert synergistic effects. It would be particularly beneficial for the elderly or the patients with preexisting medical conditions that increase their susceptibility to influenza. 

Moreover, NIR-AVPIT would also exert an indirect antiviral effect by activating host immunity similar to that activated by NIR-PIT. Previous reports have shown that NIR-PIT induces immunogenic cell death, which, unlike apoptosis, occurs rapidly [9,43,44]. Molecules released by cell disruption, particularly HMGB1, are key factors in damage-associated molecular patterns (DAMPs) [45,46], suggesting that NIR-PIT and NIR-AVPIT may mediate DAMP-induced innate immune responses in vivo and promote antigen-specific immunity by targeting pathogenic viruses and virus-infected cells. NIR-AVPIT suppresses viral replication in the early stages of infection, during which time host antiviral immune mechanisms are expected to be enhanced. This study was conducted using a cell culture model and further in vivo studies are required to confirm the effects of NIR-AVPIT on immunity. 

For the practical application of NIR-AVPIT in viral infections, optimal conditions for NIR-AVPIT in the early stages of infection must be established. It was reported that initiating oseltamivir treatment within 48 h of symptom onset decreased mortality [47]. Considering that the effective window of oseltamivir is 48 h, it would be reasonable to evaluate the optimal timing of NIR-AVPIT administration in clinical trials starting from 48 h after symptom onset. In detecting early influenza virus infections, the sensitivity of examinations is an important factor. For diagnosis of influenza, rapid antigen testing kits based on immunochromatography are widely used; however, the sensitivity of these kits is not comparable to PCR for samples with low viral loads [48]. Therefore, the development of novel kits with a higher sensitivity would contribute to earlier diagnosis of influenza. Our data suggested that multiple administration of NIR-AVPIT would be more effective than a single treatment. Designing a device that enables NIR-AVPIT to be carried out at home would facilitate repeated applications.

We evaluated NIR-AVPIT as a novel therapy against viral infections using FluV-virions and FluV-infected cells. In fact, several attempts to control viral infections by photoimmunotherapy with NIR have already been reported. However, they were intended to treat chronic retroviral infection with HIV- or HTLV-1-infected leukemic cells [14,15]. Although there has been a study using SARS-CoV-2 [16], which is an acute infection, no specific studies have been reported to analyze the relationship between time after infection and antiviral efficacy. 

In the present study, we showed that the combination of IR700-conjugated antibody and NIR irradiation effectively eliminated FluV virions and FluV-infected cells, indicating that the scope of this method extends beyond cancers to various types of pathogens [14,15,16]. NIR-AVPIT can be a fast and efficient solution against a wide range of viruses and is expected to contribute to mitigating the severity of viral infections during future pandemics. 

## Figures and Tables

**Figure 1 pharmaceutics-17-00173-f001:**
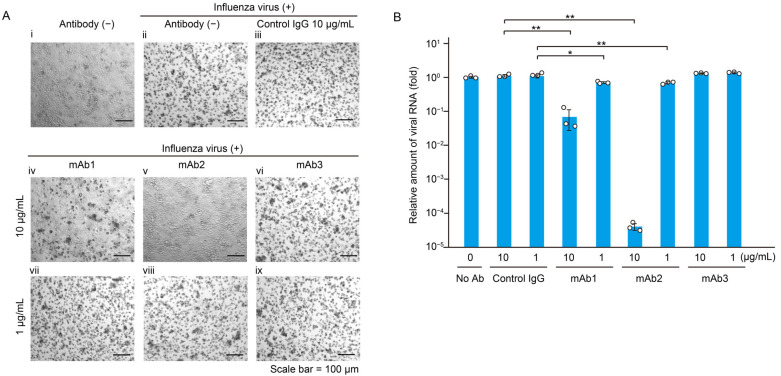
Selection of anti-HA antibody without neutralizing activity. (**A**) Neutralization activity was assessed for the three monoclonal antibodies (mAb1, mAb2, and mAb3) based on their cytopathic effect (CPE). The antibody concentrations tested were 1 µg/mL and 10 µg/mL. Cell morphology was analyzed by optical microscopy. Scale bar = 100 µm. (i) no virus/no antibody, (ii) FluV/no antibody, (iii) FluV/control IgG (10 µg/mL), (iv) FluV/mAb1 (10 µg/mL), (v) FluV/mAb2 (10 µg/mL), (vi) FluV/mAb3 (10 µg/mL), (vii) FluV/mAb1 (1 µg/mL), (viii) FluV/mAb2 (1 µg/mL), (ix) FluV/mAb3 (1 µg/mL). (**B**) The amounts of viral RNA propagated in the culture of MDCK cells during 48 h incubation. The relative viral RNA quantities were measured by RT-qPCR. The data are expressed as the mean  ± SD from three independent experiments. Asterisks indicate statistically significant differences when compared with the control cells. * *p* < 0.05, ** *p* < 0.01.

**Figure 2 pharmaceutics-17-00173-f002:**
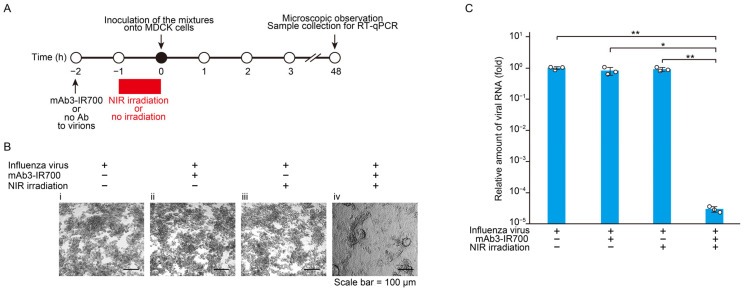
Direct inactivation of FluV particles by NIR-AVPIT. (**A**) Schematic diagram of the experiment used to evaluate the direct effect of NIR-AVPIT on FluV virions. (**B**) Representative images of CPE in MDCK cells 48 h after FluV inoculation. Scale bar = 100 µm. (i) no antibody/no irradiation, (ii) mAb3-IR700/no irradiation, (iii) no antibody/NIR irradiation, (iv) mAb3-IR700/NIR irradiation. (**C**) The amounts of viral RNA amplified in the culture of MDCK cells during 48 h incubation. The relative viral RNA quantities were measured by RT-qPCR. The experiments were repeated independently at least three times with good reproducibility. The data are shown as the mean  ± SD. Asterisks indicate statistically significant differences when compared with the control cells. * *p* < 0.05, ** *p* < 0.01.

**Figure 3 pharmaceutics-17-00173-f003:**
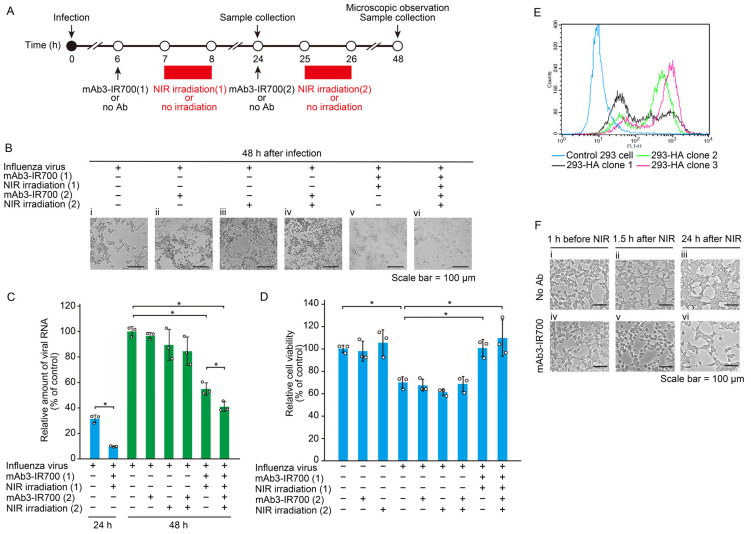
Effect of NIR-AVPIT on virus infected cells and HA-expressing cells. (**A**) Schematic diagram of the experiments to evaluate the effect of single or double NIR-AVPIT on FluV-infected cells. (**B**) Microscopic observation of MDCK cells with or without NIR-AVPIT 48 h after virus inoculation. FluV-infected cells were treated with single (at 7 or 25 hpi) or double (at 7 and 25 hpi) NIR-AVPIT. (i) no antibody/no irradiation, (ii) mAb3-IR700 (24 hpi)/no irradiation, (iii) no antibody/NIR irradiation (25 hpi), (iv) mAb3-IR700 (24 hpi)/NIR irradiation (25 hpi), (v) mAb3-IR700 (6 hpi)/NIR irradiation (7 hpi), (vi) mAb3-IR700 (6 and 24 hpi)/NIR irradiation (7 and 25 hpi). (**C**) The amounts of viral RNA produced from the infected MDCK cells with or without NIR-AVPIT at 24 or 48 hpi. The relative viral RNA quantities were measured by RT-qPCR. These data were obtained from three independent experiments and are presented as the mean ± SD. Asterisks indicate statistically significant differences with *p* < 0.05. (**D**) The viability of FluV-infected cells with or without NIR-AVPIT treatment. At 48 hpi, the viability of MDCK cells was evaluated by MTS assay. Results are shown as the mean ± SD from three independent experiments. Asterisks indicate statistically significant differences with * *p* < 0.05. (**E**) Selection of HA-expressing HEK293 clones by FACS analysis. Blue, control 293 cells (no HA expression); black, 293-HA clone 1; green, 293-HA clone 2; and magenta, 293-HA clone 3. The expression level of HA is the highest in 293-HA clone 3. (**F**) Representative micrographs of 293-HA clone 3 cells with or without NIR-AVPIT. Cell morphology was observed 1 h before NIR treatment and 1.5 h and 24 h after NIR treatment. (i) no antibody/1 h before NIR, (ii) no antibody/1.5 h after NIR, (iii) no antibody/24 h after NIR, (iv) mAb3-IR700/1 h before NIR, (v) mAb3-IR700/1.5 h after NIR, (vi) mAb3-IR700/24 h after NIR.

## Data Availability

Data are contained within the article.

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
