# Peer review of "Targeted Elimination of Influenza Virus and Infected Cells with Near-Infrared Antiviral Photoimmunotherapy (NIR-AVPIT)"

_pharmaceutics, 2025, doi:10.3390/pharmaceutics17020173_

Round 1
Reviewer 1 Report
Comments and Suggestions for Authors
The abstract effectively summarizes the study but could benefit from including specific quantitative results to highlight the treatment's efficacy.
Briefly discuss in the introduction section previous applications of photoimmunotherapy in viral infections beyond cancer to provide a broader context for the novelty of NIR-AVPIT in this study.
Elaborate on strategies to optimize the timing of NIR-AVPIT in a clinical context. Discuss the feasibility of detecting infections early enough for effective treatment and how repeated treatments could be managed.
Expand the discussion to compare NIR-AVPIT with existing antiviral therapies, highlighting advantages and potential limitations. Consider discussing synergistic effects if used in combination with standard treatments. Include a section discussing the specificity of mAb3-IR700 for infected cells versus healthy cells. Provide evidence or references that support the minimal adverse effects of NIR-PIT in similar contexts.
Please reduce the percentage match of the manuscript, which is currently 30%.
Author Response
Comment 1:
The abstract effectively summarizes the study but could benefit from including specific quantitative results to highlight the treatment's efficacy.
Response 1:
Thank you for your valuable feedback. In response to your comment, we have included the quantitative results in the abstract (lines 23-24).
Comment 2:
Briefly discuss in the introduction section previous applications of photoimmunotherapy in viral infections beyond cancer to provide a broader context for the novelty of NIR-AVPIT in this study.
Response 2:
Following your constructive comment, we have added a brief discussion about photoimmunotherapy against viral infections to the introduction section (lines 60-62).
Comment 3:
Elaborate on strategies to optimize the timing of NIR-AVPIT in a clinical context. Discuss the feasibility of detecting infections early enough for effective treatment and how repeated treatments could be managed.
Response 3:
Thank you for your helpful comments to improve our manuscript. According to your advice, we have discussed the strategies to optimize the timing of NIR-AVPIT, the feasibility of detecting early infections, and the management of repeated treatments (lines 360-372).
Comment 4:
Expand the discussion to compare NIR-AVPIT with existing antiviral therapies, highlighting advantages and potential limitations. Consider discussing synergistic effects if used in combination with standard treatments. Include a section discussing the specificity of mAb3-IR700 for infected cells versus healthy cells. Provide evidence or references that support the minimal adverse effects of NIR-PIT in similar contexts.
Response 4:
We sincerely appreciate your valuable feedback. We have substantially amended the manuscript in accordance with your comments (lines 330-333, 338-346, 307-316).
Comment 5:
Please reduce the percentage match of the manuscript, which is currently 30%.
Response 5:
Thank you for your comment. We analyzed the revised manuscript with iThenticate and the percentage match was 24%. Please let me know if further modification is necessary.
Reviewer 2 Report
Comments and Suggestions for Authors
The authors have submitted the manuscript titled "Targeted elimination of influenza virus and infected cells near-infrared antiviral phototimmunotherapy (NIR-PIT). In this study, the authors evaluated the potential of NIR-PIT technique, which is commonly used in cancer treatment, with non-neutralizing binding antibody, for inactivation of the influenza virus and elimination of the infected cells in vitro.
I have given my concerns about the study below:
1. In the materials and methods and discussion section, the authors have pasted instruction for authors from the journal's webpage. These errors must be thoroughly cross-checked and corrected.
2. The authors mention that the neutralizing antibodies are often rendered ineffective due to antigenic drift. But the binding, non-neutralizing antibodies bind to the conserved portions of the HA, and are thus resistant to antigenic drift. The authors do not give even a single reference to support this.
The authors must experimentally prove their claims. They should show which part of the HA (head vs. stem, the amino acid sequence), the non-neutralizing antibody that they show is binding to.
They have only tested this against 1 influenza virus of only 1 strain (2009 H1N1 pandemic virus). They did not test this against any other H1 virus, pre-and post-2009 pandemic H1 virus, nor they have tested this in H3 or B strains of the influenza virus. This must be tested in these viruses to support their claims.
3. In the discussion section, the authors mention that the vaccines are primary treatment options for COVID19, influenza and other respiratory infection. Do they mean therapeutic vaccines? Are any of the therapeutic vaccines commonly used for the respiratory infections?
The vaccines are commonly used for prevention, not for treatment.
4. Moreover, the authors also mention the mutant strains of the virus resulting in the loss of vaccine efficacy. Similar to the first comment, these mutant strains will also lead to decreased/loss of binding of the antibodies and will also render the proposed treatment ineffective.
Also, the influenza vaccines are administered in either trivalent or quadrivalent form to provide protection against H1, H3 and B strains of the virus. Do the authors propose trivalent or quadrivalent NIR-AVPIT?
Author Response
Comment 1: The authors have submitted the manuscript titled "Targeted elimination of influenza virus and infected cells near-infrared antiviral phototimmunotherapy (NIR-PIT). In this study, the authors evaluated the potential of NIR-PIT technique, which is commonly used in cancer treatment, with non-neutralizing binding antibody, for inactivation of the influenza virus and elimination of the infected cells in vitro.
I have given my concerns about the study below:
1. In the materials and methods and discussion section, the authors have pasted instruction for authors from the journal's webpage. These errors must be thoroughly cross-checked and corrected.
Response 1: Thank you very much for pointing out the error I made. We have deleted the pasted instructions for authors.
Comment 2:
2. The authors mention that the neutralizing antibodies are often rendered ineffective due to antigenic drift. But the binding, non-neutralizing antibodies bind to the conserved portions of the HA, and are thus resistant to antigenic drift. The authors do not give even a single reference to support this.
The authors must experimentally prove their claims. They should show which part of the HA (head vs. stem, the amino acid sequence), the non-neutralizing antibody that they show is binding to.
They have only tested this against 1 influenza virus of only 1 strain (2009 H1N1 pandemic virus). They did not test this against any other H1 virus, pre-and post-2009 pandemic H1 virus, nor they have tested this in H3 or B strains of the influenza virus. This must be tested in these viruses to support their claims.
Response 2:
Thank you very much for providing such helpful and valuable comments. Based on your feedback, we substantially revised our manuscript (lines 276-280, 318-328). Unfortunately, the epitope of mAb3 has not been determined nor other strains have been tested yet. Instead of performing additional experiments, we have included the discussion about mAb3-based NIR-AVPIT. The main objective of this study is to evaluate whether NIR-AVPIT is effective against influenza virus and infected cells. The investigation of its efficacy against a broad range of viral strains is planned for future studies.
Comment 3:
3. In the discussion section, the authors mention that the vaccines are primary treatment options for COVID19, influenza and other respiratory infection. Do they mean therapeutic vaccines? Are any of the therapeutic vaccines commonly used for the respiratory infections?
The vaccines are commonly used for prevention, not for treatment.
Response 3:
Thank you for your valuable comment. As you pointed out, the vaccines against respiratory viruses are not therapeutic. To prevent potential misunderstandings among readers, we have revised the discussion section (lines 265-266).
Comment 4:
4. Moreover, the authors also mention the mutant strains of the virus resulting in the loss of vaccine efficacy. Similar to the first comment, these mutant strains will also lead to decreased/loss of binding of the antibodies and will also render the proposed treatment ineffective.
Also, the influenza vaccines are administered in either trivalent or quadrivalent form to provide protection against H1, H3 and B strains of the virus. Do the authors propose trivalent or quadrivalent NIR-AVPIT?
Response 4:
We sincerely appreciate your helpful review for improving this manuscript. In response to your valuable comments, we have made significant revisions in the Discussion part (lines 268-271, 318-328). With regard to the valency of NIR-AVPIT, it would depend on the availability of antibodies with multi-strain reactivity. If an antibody with broad reactivity across H1 and H3 can be obtained, it would allow us to reduce the valency of NIR-AVPIT. We also included this point in the revised Discussion part.